

# Technical note:
# Hydrograph separation: How physically based is recursive digital filtering?

Klaus Eckhardt

University of Applied Sciences Weihenstephan-Triesdorf, Weidenbach, 91746, Germany

*Correspondence to*: Klaus Eckhardt (klaus.eckhardt@hswt.de)

**Abstract.** Recursive digital filtering of hydrographs is a widely used method to identify the groundwater-borne portion of streamflow. In this context, a distinction is often made between physically based and non-physically based algorithms. The algorithm of Furey and Gupta (2001), for example, is counted among the former. In this paper, it is contrasted with the

algorithm of Eckhardt (2005). This algorithm represents a whole class of recursive digital filters based on the assumption that the aquifer is a linear reservoir. It is shown that the algorithm of Eckhardt (2005) is not merely a low-pass filter, but that it is largely identical to the aforementioned physically based algorithm of Furey and Gupta (2001). The algorithm of Eckhardt (2005) differs from the algorithm of Furey and Gupta (2001) only in the time delay assumed between precipitation and the exfiltration of groundwater into surface waters, and in the fact that two parameters are combined into one, $BFI_{max}$.

This parameter can thus be interpreted physically and an approach for its calculation emerges.

## 1 Introduction

Hydrograph separation attempts to identify the portion of the streamflow that originates from groundwater, the so-called baseflow. A variety of methods has been developed for this purpose. One approach is based on the consideration that a catchment can be understood as a signal converter. The precipitation is the input signal that is converted into the output

signal, streamflow. The processes that lead to the formation of baseflow on the one hand and to the formation of faster runoff components (so-called direct runoff) on the other are likely to attenuate and delay the input signal to different degrees. From this point of view, it is obvious to low-pass filter streamflow hydrographs.

This approach has been followed since Lyne and Hollick (1979) introduced the recursive digital low-pass filter to hydrology. The term "digital" refers to the fact that discrete, equidistant in time data of the streamflow are used, the processing of which

can be easily automated by using a computer. The term "recursive" refers to the fact that the signals of the preceding time steps are included in the calculation of the output signal in the current time step.

Subsequently, several such recursive digital low-pass filters have been developed. In the present publication, the filter developed by Eckhardt (2005) is considered in particular. It is usually counted among the non-physical or "purely empirical" (Healy, 2010, p. 87) methods of hydrograph separation. The apparent lack of a physical basis repeatedly raises doubts about



the justification of the recursive digital filtering: "Most hydrograph separations (apart from tracer-based separations) lack a physical basis. […] Therefore, choosing one method or the other introduces an undesirable element of uncertainty and randomness into the analysis and comparison of runoff coefficients." (Blume et al., 2007), "The digital filter methods have no physical meaning" (Kang et al. 2022). However, without a physically meaningful interpretation, it becomes impossible to objectively determine the parameters of the filter algorithms: "parameters used in the RDF [recursive digital filtering]

method are often determined arbitrarily, resulting in high uncertainty of the estimated baseflow rate." (Zhang et al., 2013), "quantitative results of the filtering change with the value of the parameters. Although the shape of the hydrograph separation can be visually consistent with the conceptualisation of a hydrograph separation, it is basically impossible to draw any conclusion from it." (Pelletier and Andréassian, 2020), "To accurately separate the baseflow from streamflow with the digital filter methods, appropriate filter parameters must be estimated by trial and error, which act as a difficulty or limitation

on their use." (Kang et al., 2022).

Is this criticism justified? Does the widespread recursive digital filtering, especially that with Eckhardt's algorithm, really lack a physical, hydrologically plausible explanation and does the choice of parameter values remain arbitrary?

In order to shed light on the answers to these questions, Eckhardt's filter is compared below with the algorithm of Furey and Gupta (2001). The latter has been developed explicitly from hydrological principles. Its developers therefore - rightly -

describe it as physically based and emphasise the difference to the previously mentioned low-pass filters: "Unlike other filters, our filter is not founded on the assumption that base flow and overland flow are the low- and high-frequency components of streamflow, respectively". The analysis shows that there is nevertheless a close relationship between the Eckhardt (2005) and Furey and Gupta (2001) filters and thus provides a clue as to how the parameter $BFI_{max}$ of the recursive digital filter of Eckhardt (2005) can be physically interpreted and determined.

**2 The two separation methods**

**2.1 The method of Eckhardt (2005)**

The equation of this low-pass filter is

$$b_k = \frac{(1 - BFI_{max})\, a\, b_{k-1} + (1-a)BFI_{max}\, y_k}{1 - a\, BFI_{max}}, \tag{1}$$

where $b$ is the baseflow, $y$ is the streamflow, $k$ is the time step number, and $a$ and $BFI_{max}$ are parameters whose values must

be set before applying the filter. Equation (1) is subject to the condition $b_k \leq y_k$, that is, if, which is mathematically possible, $b_k > y_k$ results, $b_k = y_k$ is set.

Even though the filter of Eckhardt (2005) is contrasted here with the filter of Furey and Gupta (2001), which is explicitly described as physically based, it is nevertheless also itself based on plausible assumptions:





(a) The information about the base flow $b_k$ of the current time step $k$ lies in the base flow $b_{k-1}$ of the preceding time step $k$-1

and in the total streamflow $y_k$ of the current time step:

$$b_k = A \, b_{k-1} + B \, y_k \tag{2}$$

(Eckhardt, 2005, Eq. (8)).

(b) The aquifer is a linear reservoir, i.e. the discharge from the aquifer is proportional to the amount of water stored in it. Without further knowledge about the physical properties of the aquifer, this is the most obvious approach. The filter

parameter $a$ corresponds to the so-called recession constant of the reservoir, which can be derived from the streamflow data as described in Eckhardt (2008).

(c) The algorithm of Lyne and Hollick (1979) has been criticised as hydrologically implausible, since it shows a constant streamflow $y$ or baseflow $b$, respectively, when direct runoff $y - b$ has ceased (Chapman, 1991). Equation (1) does not have this disadvantage: From $y_k - b_k = 0$ or $y_k = b_k$ follows

$y_k = \frac{(1 - BFI_{max}) \, a \, b_{k-1} + (1-a) BFI_{max} \, y_k}{1 - a \, BFI_{max}}$.

This equation can be simplified to $y_k = a \, b_{k-1}$ or, since in this situation the streamflow consists entirely of baseflow,

$$b_k = a \, b_{k-1}. \tag{3}$$

This is exactly the equation that describes the exponential decrease in runoff from a linear reservoir.

(d) The second filter parameter $BFI_{max}$ is the maximum value of the baseflow index (the long-term ratio of baseflow to total

streamflow) that can be calculated with the filter algorithm. This maximum value is less than 1. This too is plausible. A catchment with a baseflow index of 1, i.e. a catchment without direct runoff, would have to have a soil with an extremely high infiltration and storage capacity and/or would have to be flat. In such an area, there would be no watercourse at all whose baseflow index could be determined.

The calculation with Eckhardt's algorithm requires streamflow data and the values of two parameters, with the streamflow

data allowing one of the two parameters, the recession constant $a$, to be determined. How uncertainties in the two filter parameters affect the resulting baseflow index can be calculated as Eckhardt (2012) has shown.

**2.2 The method of Furey and Gupta (2001)**

Furey and Gupta formulated their filter algorithm as

$$\bar{Q}_{B,j} = (1 - \gamma)\bar{Q}_{B,j-1} + \gamma \, \frac{c_3}{c_1} \, (\bar{Y}_{B,j-d-1} - \bar{Q}_{B,j-d-1}) \tag{4}$$

(Furey and Gupta, 2001, Eq. (22)). $\bar{Q}_{B,j}$ is the baseflow at time step $j$, $1 - \gamma$ is the recession constant, $c_1$ and $c_3$ are the proportions of precipitation that become overland flow and groundwater recharge, $\bar{Y}_{B,j-d-1}$ is the streamflow at time step $j -$





$d - 1$, and $d$ is the delay between precipitation and groundwater recharge. Using the same symbolic designation for time step number, recession constant, baseflow, and streamflow as in Eq. (1), Eq. (4) can also be written as

$$b_k = a\, b_{k-1} + (1 - a)\, \frac{c_3}{c_1}\, (y_{k-d-1} - b_{k-d-1}).$$  (5)

The calculation of the base flow according to Furey and Gupta (2001) requires streamflow and precipitation data and the values of four parameters: $a$, $c_1$, $c_3$, and $d$. Precipitation is needed for the derivation of the values of $c_1$ and $c_3$. How $d$ can be estimated remains open.

## 2.3 The relation between the two algorithms

In deriving their filter equation, Furey and Gupta (2001) assume that the baseflow in the current time step is a function of
baseflow and groundwater recharge one time step in the past (their Eq. (10)). Further, they assume that the groundwater recharge is delayed by $d$ time steps compared to precipitation (their Eq. (11)). Hence, the index $j - d - 1$ in Eq. (4) and Eq. (5).

If instead it is assumed that baseflow occurs in the same time step as groundwater recharge and groundwater recharge is not delayed to precipitation, in other words, if it is assumed that the delay between precipitation and baseflow is smaller than one
time step ($d - 1 = 0$), then Eq. (5) is

$$b_k = a\, b_{k-1} + (1 - a)\, \frac{c_3}{c_1}\, (y_k - b_k).$$  (6)

This equation can be transformed to

$$b_k = \frac{a}{1 + (1-a)\frac{c_3}{c_1}}\, b_{k-1} + \frac{(1-a)\frac{c_3}{c_1}}{1 + (1-a)\frac{c_3}{c_1}}\, y_k.$$  (7)

Equation (7) corresponds in principle to Eq. (2), which in turn is the basis of Eckhardt's algorithm. The comparison of Eq.
(7) and Eq. (1), or more precisely the comparison of the coefficients of $b_{k-1}$ and $y_k$ in both equations, yields

$$\frac{1}{1 + (1-a)\frac{c_3}{c_1}} = \frac{1 - BFI_{max}}{1 - a\, BFI_{max}}$$  (8)

and

$$\frac{\frac{c_3}{c_1}}{1 + (1-a)\frac{c_3}{c_1}} = \frac{BFI_{max}}{1 - a\, BFI_{max}}.$$  (9)

The solution of this system of equations results in

$$BFI_{max} = \frac{c_3}{c_1 + c_3}.$$  (10)





In other words, a single assumption, namely that baseflow still begins at the same time step as precipitation, is sufficient to transform the algorithm of Furey and Gupta (2001) into the algorithm of Eckhardt (2005), where the relation (10) holds.

## 3 Discussion

Eckhardt's algorithm represents a whole class of recursive digital filters that only differ by the value of $BFI_{max}$. These are the
filters that are based on the assumption of the aquifer being a linear reservoir and that are constructed according to Eq. (2). For example, setting $BFI_{max}$, = 0.5 yields the filter of Chapman and Maxwell (1996). Do these filter algorithms lack a physical basis? Section 2 should have made it clear that this is not the case. The algorithm of Eckhardt (2005) differs from the algorithm of Furey and Gupta (2001) only in the time delay assumed between precipitation and the exfiltration of groundwater into surface waters, and in the fact that two parameters, $c_1$ and $c_3$, are combined into one, $BFI_{max}$.

**3.1 Time delay**

The assumption that there is no major time lag between precipitation and the exfiltration of groundwater into surface waters is supported by the separation of hydrographs using isotope tracers. This now decades-old technique has shown time and again that there is a rapid release of so-called pre-event water into streams after precipitation (reviews: Buttle, 1994; Klaus and McDonnell, 2013).

**3.2 Model parameters**

$c_1$ is the ratio of overland flow to precipitation, $c_3$ the ratio of groundwater recharge to precipitation. Furey and Gupta (2001) propose a method to determine $c_1$ and $c_3$ using additional precipitation data. $BFI_{max}$ could then be calculated with Eq. (10). $BFI_{max}$ could also be determined in another way. If the fraction on the right-hand side of Eq. (10) is expanded with the precipitation, the result is

$$BFI_{max} = \frac{\text{groundwater recharge}}{\text{overland flow + groundwater recharge}}.$$

If one assumes approximately that (a) there is no inflow or outflow of groundwater below the surface boundaries of the catchment, and (b) there is no evapotranspiration from groundwater or surface waters, then the sum of overland flow and groundwater recharge corresponds to the streamflow:

$$BFI_{max} \approx \frac{\text{groundwater recharge}}{\text{streamflow}}. \tag{11}$$

Streamflow is given. Consequently, "only" a method for estimating mean groundwater recharge is needed to approximate $BFI_{max}$. This is plausible. The parameter $BFI_{max}$ was introduced as the maximum value of the baseflow index that can be calculated. And the baseflow can at most correspond to the groundwater recharge.



## 4 Conclusions

The recursive digital filter of Eckhardt (2005) largely coincides with the physically based algorithm of Furey and Gupta (2001). As Eckhardt (2005) has pointed out, his filter is identical with the filter of Boughton (1993) and passes for different values of the parameter $BFI_{max}$ into one-parameter filters like the one of Chapman and Maxwell (1996). Thus, the question posed in the title of this paper can justifiably be answered for a whole family of recursive digital filters with: Yes, they are physically based.

The preceding considerations also suggest a way in which the parameter $BFI_{max}$ of Eckhardt's filter could be determined
objectively, namely via groundwater recharge. Since the results of Eckhardt's filter are less sensitive to the parameter $BFI_{max}$ than to the parameter $a$ (Eckhardt, 2012), the estimate for $BFI_{max}$ would not even have to be particularly accurate. Even an uncertainty of up to almost 40 % would probably only lead to an uncertainty of less than 10 % in the calculated baseflow index.

### Competing interests

The author declares that he has no conflict of interest.

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
