# Peer review of "Technical note: How physically based is hydrograph separation by recursive digital filtering?"

_Hydrology and Earth System Sciences, 2022_

## Referee Comment (RC2)

**Review of the manuscript HESS-2022-186**

The technical node entitled *"Hydrograph separation: How physically based is recursive digital filtering?"* brings some clarifications on the physical basis of recessive digital filters and more specifically to the Eckhardt (2005) recursive digital filter. More importantly the technical note provide one approximation of the so-called $BFI_{max}$ parameter that is needed to apply the filter. The paper is well written and really clear from the beginning to the end. I just miss some clarifications and improvements that could facilitate the reading of this paper for a more general audience. I suggest accepting the paper with the following minor comments.

**Generic comments**

1. I feel like the introduction is oriented toward advanced users of recursive digital filters and especially the Eckhardt (2005) filter. I would appreciate having some examples of recent applications of the filter to underline how it is used in different contexts.

2. In section 3.1, I would rather prefer a discussion about the consequences of such an assumption (i.e., the short time delay between recharge and baseflow) and thus to what extends (hydrogeological contexts) the application of the recursive digital filter is restricted in respect to the physical meaning developed in the paper.

3. In section 3.2, equation 11 is given to provide one approximation of $BFI_{max}$. Groundwater recharge is needed to estimate the parameter. This makes sense since, for low storativity aquifers, baseflow is highly correlated to recharge so estimation of recharge would provide a good approximation of baseflow. However, I feel like using groundwater recharge to estimate such a parameter is just moving the problem to another problem. But I see the benefit of this approximation to validate $BFI_{max}$ estimates or to provide a first estimation of this parameter in a given context.

**Specific comments**

Line : 21-22 : Not clear to me what the author wants to say with this sentence. Please clarify

Equation (2) : Although A and B are described in Eckhardt 2005, might be good here to recall their meanings. This will prevent readers to read the original publication to follow the reflection.

Line 147: I miss some explications on how the author stands that a 40% recharge error will not lead to more than 10% of uncertainty in baseflow.

---

## Author Comment (AC4)

HESS-2022-186
Reply to the comments of referee 2

**Generic comments**

1   Lines 27 to 29 could be reworded to:

Several such recursive digital low-pass filters were subsequently presented. In the following, the filter developed by Eckhardt (2005) is considered in particular. It is now one of the established methods of hydrograph separation, for example it is part of the U.S. Geological Survey Hydrologic Toolbox (Barlow et al., 2022).
The "Eckhardt filter", as it is oftentimes called, is usually counted among the non-physical or "purely empirical" (Healy, 2010, p. 87) methods of hydrograph separation.

2   As I already wrote in my reply to RC1, I intend to reword section 3.1 as follows:

Furey and Gupta (2001) introduced the parameter $d$ in Eq. (5) as the number of time steps between precipitation and groundwater recharge. A sensitivity analysis they conducted showed that the filter performance was "relatively insensitive to changes in $d$" so that $d = 0$ seemed to be an acceptable choice. Furthermore, when using Eq. (1), it is assumed that not only the groundwater recharge but also the generation of baseflow still occurs in the same time step as precipitation. When assessing these prerequisites, two aspects should be considered:
(1) The streamflow component calculated with Eq. (1) is usually likely to consist not only of groundwater, but also of transient water sources, including interflow (Cartwright et al., 2014; Yang et al., 2021).
(2) In this publication, the algorithm of Eckhardt (2005) is compared to the model ideas of Furey and Gupta (2001) on the formation of baseflow. It is not compared to the reality. If the baseflow calculated with Eq. (1) occurs in Furey and Gupta's model world at the same time step as precipitation, this does not necessarily mean that it also corresponds to a runoff component in the real world that occurs without a relevant time lag to precipitation.

3   It is right that "using groundwater recharge to estimate such a parameter is just moving the problem to another problem". I indicated this by putting quotation marks around the word 'only' in the phrase "Consequently, "only" a method for estimating mean groundwater recharge is needed to approximate $BFI_{max}$." (lines 135 - 136).

**Specific comments**

•   Lines 21-22: As I already wrote in my reply to CC2, I intend to reword the first paragraph of section 1 as follows:

A catchment can be understood as a signal converter. The precipitation is the input signal that is converted into the output signal, streamflow. In the course of this signal conversion, the water takes different paths through the catchment and is subject to different hydrological processes. This results in streamflow components that are attenuated and delayed to varying degrees compared to the input signal, the precipitation. Usually, two components are distinguished: on the one hand, the so-called baseflow as a low-frequency signal component and, on the other hand, higher-frequency signal components that are generated more quickly and less attenuated in response to precipitation events, the so-called direct runoff. From this idea, it is obvious to low-pass filter streamflow hydrographs to identify these components.

•   Equation (2): I intend to add the following to lines 59 to 62:

(a) The information about the baseflow $b_k$ of the current time step $k$ lies in the baseflow $b_{k-1}$ of the preceding time step $k-1$ and in the total streamflow $y_k$ of the current time step:

$$b_k = A\,b_{k-1} + B\,y_k \qquad\qquad (2)$$

with parameters $A$ and $B$ that are functions of the filter parameter $a$ and for which $A > 0$ and $B > 0$ is assumed (Eckhardt, 2005, Eq. (8)).

•   Line 147: I would like to explain the mentioned estimate by adding the following to the text:

Since the results of Eckhardt's filter are less sensitive to the parameter $BFI_{max}$ than to the parameter *a* (Eckhardt, 2012), the estimate for $BFI_{max}$ would not even have to be particularly accurate. The sensitivity of the baseflow index $BFI$ to the parameter $BFI_{max}$ can be described by the sensitivity index

$$S(BFI|BFI_{max}) = \frac{(a-1)(a\,BFI-1)}{(1-a\,BFI_{max})^2}\frac{BFI_{max}}{BFI} \tag{12}$$

(Eckhardt, 2012, Eq. (15)). For sixty perennial streams with porous aquifers, Eckhardt (2012) has found a mean sensitivity index of 0.26. That is, a relative error of X percent in $BFI_{max}$ would result in a relative error of 0.26 times X percent in $BFI$. Thus, even if $BFI_{max}$ had an uncertainty of up to about 40 %, this would probably produce an uncertainty of less than 10 % in the calculated baseflow index.

- In the references it would be necessary to add:

Barlow, P. M., McHugh, A. R., Kiang, J. E., Zhai, T., Hummel, P., Duda, P., and Hinz, S.: U.S. Geological Survey Hydrologic Toolbox - A graphical and mapping interface for analysis of hydrologic data: U.S. Geological Survey Techniques and Methods, book 4, chap. D3, 23 p., https://doi.org/10.3133/tm4D3, 2022.

---

## Author Response (AR1)

Dear Dr. Bogaard,

thank you and all the reviewers for the comments. I have responded in detail during the discussion. Are these answers sufficient for you to justify my changes in the text? The main points are:

- Following Keith Beven's comment, the terms "groundwater" and "pre-event water" are now largely omitted.
- Ambiguities as to how the time delay between precipitation and baseflow is to be understood in the two algorithms should be eliminated.
- It is explicitly pointed out that this is a comparison of two algorithms and that no further statements are made about the actual reproduction of hydrological processes.

Please let me know if additional explanations are necessary.

I have one additional request: Since my contribution was classified as a "technical note", the previous version has resulted in two words followed by a colon appearing twice in succession in the title: "Technical note: Hydrograph separation: How physically based is recursive digital filtering?". Do you agree if I therefore rephrase the title as "Technical note: How physically based is hydrograph separation by recursive digital filtering?".

Sincerely,

Klaus Eckhardt